# Temporal Relationship of Extraintestinal Manifestations in Inflammatory Bowel Disease

**DOI:** 10.3390/jcm10245984

**Published:** 2021-12-20

**Authors:** Istvan Fedor, Eva Zold, Zsolt Barta

**Affiliations:** 1Department of Clinical Immunology, Institute of Internal Medicine, Faculty of Medicine, University of Debrecen, Moricz Zs. krt. 22, H-4032 Debrecen, Hungary; zold.eva@med.unideb.hu; 2Department of Public Health and Epidemiology, Faculty of Medicine, University of Debrecen, Kassai u. 26, H-4012 Debrecen, Hungary; 3Petranyi Gyula Doctoral School of Clinical Immunology and Allergology, University of Debrecen, Nagyerdei krt. 98, H-4032 Debrecen, Hungary; barta@med.unideb.hu; 4GI Unit, Department of Infectology, Faculty of Medicine, University of Debrecen, Bartok B. u. 2-26, H-4031 Debrecen, Hungary

**Keywords:** inflammatory bowel disease, Crohn’s disease, ulcerative colitis, extraintestinal manifestations

## Abstract

Objectives: Thus far, few attempted to characterize the temporal onset of extraintestinal manifestations (EIM) in inflammatory bowel diseases (IBD). We sought to determine the time of onset of these findings in a patient cohort with IBD. Methods: We reviewed the electronic health records of 508 IBD patients (303 CD, 205 UC) and summarized general patient characteristics and the temporal relationship and order of presentation of extraintestinal manifestations. Results: CD patients were younger at diagnosis. CD patients with ileocolonic involvement (L3) were younger, and UC patients with pancolitis (E3) were slightly younger at diagnosis. A total of 127 out of 303 (41.91%) CD and 81 out of 205 (39.51%) UC patients had EIMs (*p* = 0.5898). Some patients presented with EIMs before the diagnosis of IBD (9.45% of Crohn’s disease and 17.28% of ulcerative colitis patients with EIMs, respectively). Of these, seven cases (four in CD and three in UC) were visible by inspection of the patients (either dermatologic or ocular findings). The diagnosis of IBD and extraintestinal symptoms often occurred within a year (22.83% of CD and 16.04% of UC patients). Typically, the diagnosis of the first extraintestinal symptoms happened after the onset of bowel disease (+4.3 (±6.3) years, range: 10 years before to 30 years after in Crohn’s disease and +3.8 (±10) years, range: 24 years before to 30 years after) in ulcerative colitis. UC patients with pancolitis (E3) usually had EIMs earlier in the disease course and displayed EIMs more frequently before IBD diagnosis. Furthermore, patients with pancolitis developed EIMs more frequently than other sub-groups. Conclusion: Extraintestinal manifestations in inflammatory bowel diseases can present at any time, relative to the bowel symptoms. In cases, the presence of a characteristic EIM might be a harbinger of the development of IBD.

## 1. Introduction

Inflammatory bowel diseases (IBD: Crohn’s disease (CD) and ulcerative colitis (UC)) are chronic immune-inflammatory disorders, with predominantly gastrointestinal complaints [1,2]. CD can affect any part of the gastrointestinal tract; UC is restricted to the large bowel. Even though IBD patients mostly present with gastrointestinal complaints, they are known to display symptoms beyond the bowels [2,3,4,5]. The extraintestinal phenomena might have a secondary origin to chronic inflammation, medication, and nutrient malabsorption (these are known as extraintestinal complications—EC). On the other hand, extraintestinal manifestations (EIMs) are not a direct consequence of medications or nutritional status and have an immune-inflammatory origin. Other organs frequently involved include the skin, joints, eyes, and the hepatobiliary–pancreatic systems [2,6,7,8,9,10]. Some of the extraintestinal manifestations (EIMs) are known to be associated with disease relapses, whereas others have a course independent of IBD.

Thus far, only Vavricka et al., assessed the temporal relationship of these conditions in adult IBD cases [5]. While it is accepted that the diagnosis of extraintestinal symptoms might either precede or follow that of IBD (and frequently, the conditions are diagnosed simultaneously), very few have attempted to quantify the temporal appearance of EIMs.

In our study, we sought to characterize patients with inflammatory bowel disease in Northeastern Hungary, treated by a single outpatient clinic, at the Clinical Center, University of Debrecen.

## 2. Materials and Methods

We retrospectively searched the previous medical documentation of 508 (303 with CD, and 205 with UC) patients with inflammatory bowel disease treated by a single outpatient clinic at the University of Debrecen Clinical Center. The collected data included gastroenterology visits from 2008 onwards until 2021. We included 133 males and 170 females with CD, whereas 89 males and 116 females had UC. Thus, the total number of male and female participants was 222 and 286, respectively. All of the patients had their condition confirmed via endoscopy and biopsy specimens.

We collected the available data in an Excel spreadsheet (Excel 2019, Microsoft Corporation, Redmond, Washington, DC, USA). The extracted data were analyzed with Medcalc Software (Medcalc Software Ltd. Version 20.014, Ostend, Belgium). Differences between the groups (age of onset, extraintestinal comorbidities) were calculated. The comparison of proportions was completed through the chi-square test, as recommended for small sample sizes by Campbell [11]. An independent sample (unpaired) two-tailed *t*-test was used to compare the two groups’ age of diagnosis. The recorded data included the age of diagnosis, the number of extraintestinal manifestations the patients had, and the temporal relationship (in years, compared with the onset of IBD) of these extraintestinal findings. In those with more than one extraintestinal manifestation, we also sought to organize these according to the sequence in which they appeared. Furthermore, Crohn’s disease patients were classified according to the Montréal system (to assess the length of inflamed bowel (L) and the behavior of their disease (B)). UC cases were evaluated by the maximum extent of colonic inflammation (Montréal E subtype). We sought whether there were any characteristics of different Montréal E-groups (age of disease onset, presence of EIMs, and their onset in relation to bowel disease). Comparison of three or more groups was carried out via ANOVA, and the interpretation of the results always considered the effect of the small size of our study cohorts.

## 3. Results

### 3.1. General Characterization of Crohn’s Disease Patients

Crohn’s disease patients (*n* = 303, 133 males, 170 females) were classified according to the Montréal system (age of disease onset (A), the length of inflamed bowel mucosa (L), disease behavior (B), and whether patients had perianal disease (p)). Montréal classification data are listed in Table 1. The average age of diagnosis in CD patients was 32.39 years (±13.23 years; range 7–79 years). Comparing the two patient groups, CD patients were younger at diagnosis (32.39 ± 12.23 years (CD); 35.02 ± 13.82 years (UC); difference = 2.63 years, *p* = 0.0313—data not shown). The most typical “age” category according to the Montréal system was A2 (199 patients (65.68% of all CD cases), diagnosed between 17 and 40 years of age).

### 3.2. Affected Bowel Segments in Crohn’s Disease and Ulcerative Colitis—Relationship to Age at Diagnosis

The extent (length—L) of inflamed sections was also recorded. As we searched medical records from 2008 to 2021, the designated Montréal L and B subtypes reflected the disease course. The upper intestinal subtype, L4, was rare (only six patients—2% total—were reported to have CD confined to upper intestinal segments). L4 cases are probably being underreported, as upper endoscopy is not routinely performed during diagnostic workup [12]. Upper gastrointestinal lesions are more common in pediatric cases and are usually associated with a more severe disease course [13,14]. We would like to mention that when we were evaluating whether there was a difference between different Montréal L groups in the average age at diagnosis (Table 2), we found that the L4 subgroup displayed a tendency to be recognized in advanced age. Thus, age cannot be used as an exclusion criterion for upper gastrointestinal Crohn’s disease. Increasing awareness of upper GI involvement might cause more and more adult and senior people to be diagnosed with the L4 type of Crohn’s disease. In our patient cohort, the disease involved the colon 71.62% of the time (colonic involvement only in 89 patients (L2: 29.37%) and was ileocolonic in 128 patients (L3: 42.24%). The “archetypical” ileal-only involvement struck 70 patients (L1: 23.1%). Colonic involvement was higher than we expected, in comparison with previous literature [15]. There was a marked difference in the average age of diagnosis in the Montréal L3 subgroup: patients were younger when compared with L1 or L2 groups (ANOVA: F = 7.76; *p* = 0.001, the difference was also marked, when we compared L1 and L3 groups via chi-square test, diff = 6.24 years, *p* = 0.0005). We also included the age of diagnosis in UC with different Montréal E subtypes. For the results, please see Table 2.

### 3.3. Extraintestinal Findings in CD and UC—An Overview

The overview of extraintestinal manifestations is listed in Table 3. The breakdown of individual EIMs in the two groups (CD and UC) is displayed in Table 4. The most common finding was peripheral arthritis. Other types of musculoskeletal involvement (sacroiliitis, spondylitis) were also commonly reported.

Previous data are controversial as to whether EIMs are more common in CD or UC [6,16,17,18]. In our study group, 127 out of a total of 303 Crohn’s disease patients had at least one EIM; this number was 81 out of 205 UC cases. Thereby, 41.91% of CD and 39.51% of UC patients reported having at least one EIM. This difference between groups was only marginal (2.4% difference, *p* = 0.5898).

The difference between males and females in whom EIMs were diagnosed was marked enough to reach a statistically significant level. In total, 70 male and 138 female patients had at least one EIM (out of 222 and 286 male and female patients, respectively). Thereby, the percentage of affected patients was 31.53% in male and 48.25% in female patients (*n* = 508 patients total; difference = 16.8 %, *p* = 0.0001). The difference between genders was not as marked when we compared the proportion of males and females with more than one EIM. In total, 9.9% of male and 15.38% of female patients had two or more EIMs, and this yielded a nonsignificant difference of 5.48%, *p* = 0.0686. This prevalence of multiple extraintestinal comorbidities is slightly less than we expected. In previous reports, more EIMs were present in roughly one-fourth of patients with IBD [19,20].

Even though we sought to determine whether there was a difference between Montréal L subgroups regarding the presence of extraintestinal findings, we were unable to observe any differences (displayed in Table 5). The proportion of patients with a diagnosed EIM was similar in every Montréal L subgroup. The results were similar in UC, though there was a trend that patients with more extensive inflammation (E2 and E3 subtypes) developed EIMs slightly more frequently. Nonetheless, due to the small sample size and comparable proportions, the difference between groups was not marked.

### 3.4. Extraintestinal Manifestations Affecting Different Organ Systems—The Musculoskeletal System Was the Primary Site Involved

The most frequent extraintestinal complaints were peripheral joint inflammatory findings (pain, swelling, warmth, impaired daily functioning; reduced range of motion), as well as axial skeleton involvement (sacroiliitis and spondylitis). Though peripheral arthritides are further subdivided into two types, we did not dichotomize, mostly due to insufficient available data. Axial joint complaints were equally common in CD and UC in our samples. Axial manifestations are reported to be more prevalent in males, but we could not confirm this. In this study sample, in total, 34 females and 17 males suffered from sacroiliitis, verified via imaging studies. The number of patients with sacroiliitis was greater in cases of CD (32 patients, 20 females, 12 males, as opposed to 19 patients—14 females, 5 males—in UC), but this difference was nonremarkable. As the total number of patients was 508, of whom 51 had been diagnosed with sacroiliitis during their disease course, the prevalence of sacroiliitis was roughly 10% in our study group. This is lower, than we expected, as other studies reported greater prevalence [21,22,23,24,25]. Vertebral column involvement in the form of spondylitis or ankylosing spondylitis was seen in 25 patients total—18 females and 7 males. The ratio of affected patients was comparable in CD and UC was 4.62% of CD (14 patients), and 5.37% of UC (11 patients) cases had vertebral column involvement.

Skin lesions, specifically erythema nodosum (EN) and pyoderma gangrenosum (PG), are well recognized in IBD [26]. Previous reports found that they can either precede or follow IBD diagnosis; occasionally, they are recognized simultaneously [5,18,26,27]. In our cohort, EN seldom presented before the onset of IBD; typically, it developed during the disease course. A total of 27 patients experienced erythema nodosum along the disease course, of whom only 3 presented before bowel symptoms (data not shown). Erythema nodosum was more common in Crohn’s disease (22 cases (7.26%) as opposed to 5 (2.44%) in ulcerative colitis). It is important to note that erythema nodosum might also resolve with the interleukin 12/23 inhibitor ustekinumab, and this agent is also a therapeutic option in Crohn’s disease [28]. The remaining cases of EN were either diagnosed at the same time (*n* = 10) or after the bowel disease onset (*n* = 13). Psoriasis—though an independent disease entity from IBD—was also found in five cases: two with CD and three with UC. It is noteworthy to add that two of those patients received biologic therapy (adalimumab) for their IBD; thereby, adverse drug effects (psoriasiform eruptions) cannot be ruled out in their cases [29].

Comorbidities related to the hepatobiliary system were less prevalent. We did not register cases with liver steatosis (fatty liver) or hypertransaminasemia (elevated ALT and AST), as these conditions are common in the general population. Non-alcoholic fatty liver disease has an estimated global prevalence of 25%, this is similar to the incidences described in Hungary [30,31]. Cholelithiasis, though not considered an extraintestinal manifestation of IBD, was encountered at a similar rate in CD and UC. A total of 8.58% of CD and 6.83% of UC patients had cholelithiasis in their disease history (*p* = 0.4603; data not shown). This is contradictory to the findings reported in previously published studies [18,32].

The total number of cases with primary sclerosing cholangitis (PSC) and autoimmune hepatitis (AIH) was low in our patient groups (the latter presenting exclusively in UC). Nevertheless, the two recorded cases with autoimmune hepatitis were diagnosed in patients also having primary sclerosing cholangitis. Both cases of AIH were recorded in male patients, even though this autoimmune disease is predominantly found in females [33,34]. Furthermore, even though PSC is known to be associated mostly with ulcerative colitis (the 80% of all IBD-PSC cases present in UC), in our cohort we observed no difference between the two diseases [35]. There were four patients in both groups respectively; thereby, 1.32% of CD and 1.95% of UC patients had comorbid PSC. The groups did not differ in this regard (intergroup difference: 0.63%, *p* = 0.5761). This is also a conflicting finding with previous literature [36].

### 3.5. Temporal Order of Presentation—Extraintestinal Symptoms Might Present before IBD

We determined the proportion of patients in whom extraintestinal signs preceded the onset of IBD. We were not always able to determine the exact onset of these findings; thus, we only included cases wherein we had sufficient data on the disease course. The findings are listed in Table 6. Although most of the time the diagnosis of EIMs happened during the disease course, in a substantial group of patients, the presentation of their extraintestinal symptoms preceded the actual onset of their IBD (9.45% of CD and 17.28% of UC patients, respectively). Additionally, in cases, more than one EIM was diagnosed before the bowel disease. Four patients with Crohn’s disease and two with UC had more than one extraintestinal manifestation before the diagnosis of their IBD.

Even though certain visible-on-inspection extraintestinal findings might prove useful in assessing IBD risk, we found that these findings (dermatologic or ophthalmologic manifestations) seldom preceded the development of bowel disease. There were only seven cases total when either a dermatologic symptom or eye disease presented before IBD. Of these, three cases of erythema nodosum and one case of pyoderma gangrenosum were diagnosed in patients with Crohn’s disease, whereas there were two cases of iridocyclitis and one case of episcleritis in ulcerative colitis. Thus, Crohn’s disease patients were more inclined to develop dermatologic EIMs before their bowel symptoms. In ulcerative colitis, the preceding signs were more likely to be ophthalmologic (iridocyclitis and episcleritis). Nonetheless, the total number of cases and proportion of patients who developed EIMs before IBD was too small to draw any conclusions.

Furthermore, we sought to identify whether there were differences between different patient groups in the onset of EIMs according to the extent of their bowel disease. The overview of the results is provided in Table 7. Crohn’s disease patients from different Montréal L subgroups and ulcerative colitis patients from different Montréal E (extent) groups were studied. We found that in Crohn’s disease, the involvement of the colon might predispose patients for earlier onset of extraintestinal manifestations. The difference between the groups did not reach a level of statistical significance (one-way ANOVA revealed F = 0.278, *p* = 0.841), but this might have been overcome by a larger study group. (For the results, please see Table 6.) A similar pattern was observed in cases of UC. Though more extensive involvement often meant that patients had EIM registered earlier in the disease course, this tendency was not statistically significant (one-way ANOVA showed: F = 1.174, *p* = 0.315). We thus conclude that a larger study group would have been needed to display marked differences between different groups.

Moreover (data not shown), we found that in Crohn’s disease (when the onset of EIMs was determinable) with both ileal and colonic involvement (Montréal L3 group), nearly 15% of patients had EIMs registered before the onset of IBD. This ratio was only 8% in Montréal L1 and L2 groups. Thus, it is possible that the presence of typical IBD-associated EIMs in otherwise non-symptomatic patients might foretell that later an extensive form of Crohn’s disease will develop—involving both the colon and ileum. A similar pattern was observable in ulcerative colitis as well. A total of 5 out of 16 patients (31%) with E3 (pancolitis, extended throughout the entire colon) developed EIMs before bowel symptoms, whereas this ratio was 6/24 (25%) in the E2 (left-sided colitis) and 3/30 (10%) in the E1 (proctitis) group. Although the samples were small, there was an observable tendency that greater affected bowel length meant an earlier diagnosis of an EIM.

Another novel aspect of our results is that we attempted to organize the order of sequence in which EIMs appeared when patients suffered from more than one of these conditions. Thereby, we highlighted cases wherein more than one extraintestinal symptom was mentioned in the patient’s records. The data are summarized in Table 8. Note that the first two extraintestinal manifestations were predominantly musculoskeletal system-related. There were cases, where the joint symptoms preceded the diagnosis of IBD for more than a decade. (We also observed that whenever the axial skeleton was involved, the onset usually followed that of peripheral arthritis.) There were no observable patterns in the order in which extraintestinal findings were described, though our samples with multiple EIMs were too small to draw any conclusions in this regard. Further examination on larger patient groups would be needed to determine whether there are characteristic patterns in the development of extraintestinal manifestations.

### 3.6. Patients with Multiple EIMs

We found that having more EIMs was equally common in CD and UC. Whereas 14.52% of all Crohn’s cases had two or more EIMs, this was 10.73% in UC patients (44 patients with CD had more than one EIM, whereas this number was 22 in UC). This 3.79% difference (*p* = 0.2130) supports the idea that patients are equally prone to developing further EIMs, regardless of whether they have CD or UC. Previous studies often reported multiple EIMs to be present in Crohn’s disease [18,37]. In Crohn’s disease, we evaluated. whether different Montréal L subgroups showed differing propensity for the development of more than one EIM. We found that forms of Crohn’s disease with colonic involvement (thus, L2 and L3 subtypes) were slightly more likely to develop additional EIMs during disease course (whereas we were able to find the mention of 2 or more EIMs in the history of L1 and L4 patients in 8 cases (without colonic inflammation, 10.53% of 76 patients); in L2 and L3 subtypes, this number was 36 (16.6% of 217 patients)). Probably due to small sample size, the difference did not reach the level of statistical significance (difference: 6.07%; *p* = 0.2033).

The co-occurrence of more than one musculoskeletal impairment (clustering of more than one joint involvement) occurred in 27 patients with Crohn’s disease and 15 patients with ulcerative colitis.

## 4. Discussion

In a considerable proportion of patients, the presentation of extraintestinal symptoms predated the onset of their bowel disease. We regard this to be an important point, as certain EIMs are visible upon inspection (such as ocular and dermatologic manifestations) and might raise the importance of being vigilant about subsequent bowel symptoms later. In our patient cohort, seven patients displayed “visible-on-inspection” extraintestinal findings. Of these, every dermatologic symptom was registered in patients who later developed Crohn’s disease, and ocular symptoms were seen in those who developed ulcerative colitis.

We also aimed to determine whether there was a relationship between the affected bowel length and the presence of one or more EIM in Crohn’s disease. We found no difference between different Montréal L subgroups in this regard. We also evaluated whether there were differences in the onset of EIMs in different Montréal L subgroups, and we did observe a mild tendency that patients with colonic involvement had EIMs earlier in the disease course. In ulcerative colitis, the inflammation of the entire colon (pancolitis—Montréal E3) predisposed patients to earlier development of EIMs. Moreover, roughly one-third of patients who developed pancolitis had their EIM registered before their bowel disease. We would like to mention that as we were collecting data across more years of patient history, we included the greatest extent of their bowel disease and classified patients accordingly. We did not register the severity of symptoms, while every patient underwent remission and occasional relapses during the disease course.

In both CD and UC, the first extraintestinal symptoms were usually related to the musculoskeletal system and could affect both axial and peripheral joints. Notably, whenever more than one musculoskeletal EIM was present, patients usually developed peripheral arthritis earlier in the disease course.

The dermatologic phenomenon known as metastatic CD is probably underreported [38]. We were only able to find it in two cases in our patient cohort (data not shown). Psoriasis is a possible associated immune-mediated disease. Whenever psoriasis is present, patients are prone to develop psoriatic arthritis, further complicating the clinical picture. Therapeutic interventions in IBD and psoriasis overlap, as IL-12/23 and TNFα blockage seems to be beneficial in both conditions (psoriasis and IBD) [39,40]. Though not widely used, agents interfering with the IL-17 pathway were shown to increase the risk of development of IBD [41].

Although not considered to be an extraintestinal manifestation of IBD, conjunctivitis and dry eye was the most frequently reported ophthalmologic condition in our patient cohort, and Sjögren’s syndrome was also diagnosed in seven patients total: four with Crohn’s disease and three with ulcerative colitis. Insufficient lacrimation is also recognized in IBD (DED—“dry eye disease”—dysfunction of the lacrimal glands), thereby not every case with reduced tear production qualifies as Sjögren’s syndrome [42,43]. Moreover, Sjögren’s syndrome can be associated with joint complaints and pancreatitis [44,45]. Thereby, this could confound the clinical picture in patients.

The management of the underlying IBD is crucial in most cases with EIMs. In modern era therapeutics, biologic agents (e.g., anti-TNFα therapies, anti-integrin therapies—vedolizumab) were shown to offer benefits during the course of various EIMs [28,29,46,47,48,49].

The limitations of our paper stem from the design and execution: data were collected retrospectively and are thus prone to recall bias regarding the exact disease course of individual patients [50]. We would like to mention that the treatment of extraintestinal symptoms was not managed by gastroenterologists but rather by different professionals. We limited our record-keeping to a limited number of manifestations (listed in Table 4). We did not register cases of thromboembolic events or other cardiovascular diseases. We did not include the frequency of oral mucosal findings, even though they are well known to be associated with IBD [47]. In our cohort, we were only able to find the mention of oral aphthous ulcers in the records of three patients; therefore, we omitted these from our paper. We did not seek an association between patients’ medication and disease course. In Hungary, non-steroid anti-inflammatory agents (NSAIDs) are often over- and misused; previous studies suggest that they can exacerbate IBD [51]. Early use of proton pump inhibitors (PPIs) was also proposed for increasing subsequent IBD risk, and these agents are also widely prescribed in Hungary [52]. As several authors already characterized the most common lifestyle risk factors for the development of IBD, we did not study these in our patient cohorts. Another shortcoming of our findings is the relatively small number of cases included (303 CD, 205 UC). Therefore, future investigations on larger patient cohorts would be warranted to better characterize the clinical features of EIMs in Hungarian IBD patients.

## 5. Conclusions

In conclusion, extraintestinal manifestations are equally common in Crohn’s disease and ulcerative colitis, affecting roughly 4 in every 10 patients. Patients often experience more than one extraintestinal manifestation during the disease course. Though generally diagnosed after IBD onset, EIMs could also be present before bowel symptoms (9.45% of Crohn’s disease and 17.28% of ulcerative colitis patients developed extraintestinal conditions before their bowel disease). Several patients had extraintestinal and bowel diseases recognized within a year (22.83% of CD and 16.04% of UC patients). Typically, the diagnosis of EIMs followed that of IBD, with +4.3 (± 6.3) years in Crohn’s disease and +3.8 (±10) years in ulcerative colitis. UC patients with more extensive inflammation (E3—pancolitis) usually developed EIMs earlier in the disease course, and there was a tendency in CD that the more extensive ileocolonic (L3) form also meant a somewhat earlier registration of EIM. Patients diagnosed at an earlier age were more likely to develop pancolitis in UC. The true prevalence of upper gastrointestinal lesions in CD is probably underreported.

## Figures and Tables

**Table 1 jcm-10-05984-t001:** Characteristics of patients with Crohn’s disease patients according to Montréal classification system.

Age of Diagnosis (A)	
A1 (below 17 years of age)	17 (5.8%)
A2 (17–40 years of age)	199 (67.7%)
A3 (above 40 years)	78 (26.5%)
Extent of Affected Bowel (L)	
L1—ileal	70 (23.9%)
L2—colonic	89 (30.4%)
L3—ileocolonic	128 (43.7%)
L4—upper gastrointestinal	6 (2%)
**Disease Behavior (B)**	
B1—nonstricturing, nonpenetrating	78 (26.6%)
B2—stricturing	116 (39.6%)
B3—penetrating	99 (33.8%)
perianal disease (p) modifier	69 (23.5%)

Colonic involvement (L2 and L3) was more common than expected. We found the upper gastrointestinal type to be rare, though we probably have an underreport of total cases.

**Table 2 jcm-10-05984-t002:** Comparison of the average age at diagnosis of different Montréal L subgroups in Crohn’s disease. The extent of involved colon length and average age at diagnosis in ulcerative colitis.

Crohn’s Disease: Involved Bowel Length	No. of Patients	Average Age at Diagnosis
L1—ileal	70	34.94 (±11.8) years
L2—colonic	89	34.1 (±13.4) years
L3—ileocolonic	128	28.7 (±12) years
L4—upper GI	6	47.5 (±19.2) years
Ulcerative Colitis—Maximum Extent of Inflammation	No. of Patients	Average Age at Diagnosis
E1—ulcerative proctitis	99	36.1 (±14.55) years
E2—left-sided colitis	68	34.7 (±13.6) years
E3—pancolitis	37	32.7 (±12.2) years

Note that Crohn’s disease patients with ileocolonic involvement were younger at diagnosis. Though the upper GI involvement is regarded more prevalent in pediatric cases, in our patient cohort, senior individuals were more likely to be diagnosed with this (L4) subtype. Probably the increasing awareness of the condition will mean more adult diagnoses of this subtype. Due to the small sample size, there were no marked differences in the average age at diagnosis of different Montréal E subgroups in UC, but patients with more extensive colitis tended to be younger at disease onset (NS, one-way ANOVA: *p* = 0.436, F = 0.832).

**Table 3 jcm-10-05984-t003:** Comparison of Crohn’s disease and ulcerative colitis patients and the prevalence of extraintestinal findings.

Patients with EIM	Crohn’s Disease	Ulcerative Colitis	Total Patients
Male	44 (33%)	26 (29.2%)	70 (31.53%)
Female	83 (48.8%)	55 (47.4%)	138 (48.3%)
Patients with More Than One EIM	Crohn’s Disease	Ulcerative Colitis	Total Patients
Male	16 (12%)	6 (6.7%)	22 (9.9%)
Female	28 (16.5%)	16 (13.8%)	44 (15.38%)

Note: Females were more likely to develop extraintestinal findings, and the difference between genders reached the level of statistical significance (difference = 16.8%, *p* = 0.0001). The difference was not as marked in cases with more than one EIM (9.9% of males and 15.38% of females had more than one EIM; difference: 5.48%, *p* = 0.0686).

**Table 4 jcm-10-05984-t004:** Prevalence of different EIMs in Crohn’s disease and ulcerative colitis.

Patients	Crohn’s Disease	Ulcerative Colitis
Had at least one EIM	127 (41.91%)	81 (39.51%)
Arthritis	87 (28.71%)	52 (25.37%)
Sacroiliitis	32 (10.56%)	20 (9.76%)
Spondylitis	14 (4.62%)	11 (5.37%)
Erythema nodosum	22 (7.26%)	5 (2.44%)
Pyoderma gangrenosum	7 (2.31%)	5 (2.44%)
Iridocyclitis	4 (1.32%)	5 (2.44%)
Scleritis	2 (0.66%)	3 (1.46%)
Primary sclerosing cholangitis	4 (1.32%)	4 (1.95%)

Musculoskeletal manifestations were the most common. There were only scarce reports on iridocyclitis (anterior uveitis) and (epi)scleritis. Dermatologic manifestations were rare. Though not shown, two patients with autoimmune hepatitis (AIH) were verified in the ulcerative colitis group; both of them also had PSC and were male. Primary biliary cirrhosis—not considered to be an EIM—was also diagnosed in two cases (not shown).

**Table 5 jcm-10-05984-t005:** Affected bowel segment and presence of EIMs in patients with Crohn’s disease and ulcerative colitis.

Crohn’s Disease	
L1—ileal-only involvement with at least one EIM	30 (out of 70 patients; 42.86%)
L2—colonic-only involvement with EIM	38 (out of 89 patients; 42.7%)
L3—both ileal and colonic involvement with EIM	57 (out of 128 patients; 44.53%)
L4—upper GI tract involvement	2 (out of 6 patients; 33%)
Ulcerative Colitis	
E1—ulcerative proctitis with at least one EIM	36 (out of 99 patients; 36.3%)
E2—left-sided colitis with EIM	29 (out of 68 patients; 42.65%)
E3—extensive (pancolitis) with EIM	16 (out of 37 patients; 43.24%)
Grouped: Colonic, Small Intestinal and Colonic with Small Intestinal Involvement and EIM	
Colonic (All UC groups and Crohn L2)	119 (out of 293 patients; 40.61%)
Small intestinal (Crohn L1 and L4)	32 (out of 76 patients; 42.1%)
Crohn L3 (colonic and small intestinal)	57 (out of 128 patients; 44.53%)

The differences between different groups (L1–L3) were marginal (*p* = 0.79). The L4 subtype was so rare that one cannot draw any conclusions on the prevalence of extraintestinal findings. In ulcerative colitis, there was a tendency that patients with more extensive inflammation developed EIMs more frequently. Though there were no marked differences between subgroups (*p* = 0.46), a larger sample would have yielded significant results. Note also that there was a non-marked difference between patients, whether they had colonic, small intestinal, or both colonic and small intestinal involvement in the presence of at least one EIM.

**Table 6 jcm-10-05984-t006:** Temporal relationship of EIMs in Crohn’s disease and ulcerative colitis.

	Crohn’s Disease	Ulcerative Colitis
Average difference to IBD diagnosis	+4.3 years (SD: 6.3 years; range: 10 years before to 30 years after	+3.8 years (SD: 10 years; range: 24 years before to 30 years after)
EIM before IBD	12 (9.45%)	14 (17.28%)
EIM after IBD	69 (54.33%)	45 (55.55%)
EIM and IBD were diagnosed within a year	29 (22.83%)	13 (16.04%)

Most of the time, EIMs developed later than inflammatory bowel disease. The proportion of patients in whom EIM presented before the onset of IBD did not differ significantly in CD and UC (9.45% and 17.28%, respectively; difference: 7.83%, *p* = 0.0967).

**Table 7 jcm-10-05984-t007:** The average difference between the diagnosis of IBD and various extraintestinal manifestations in different patient groups.

Crohn’s Disease Patients with EIM—Montréal L	
L1—ileal only—25 patients	+5.12 years (±6.38 years)
L2—colonic—36 patients	+3.86 years (±4.86 years)
L3—ileocolonic—47 patients	+4.28 years (±7.26 years)
L4—upper gastrointestinal—2 patients	+2.12 years (±2.12 years)
Ulcerative Colitis Patients with EIM—Montréal E	
E1—ulcerative proctitis—30 patients	+6.1 years (±7.67 years)
E2—left-sided colitis—24 patients	+3.56 years (±10 years)
E3—pancolitis—16 patients	+0.81 years (±11.7 years)

The difference between the diagnosis of IBD and registration of EIM. Even though the difference between subgroups did not reach a significant level, there is a mild tendency in both CD and UC. In CD, patients with colonic involvement developed EIMs earlier than those with pure ileal inflammation. Unfortunately, the variance within groups was rather large, thereby making the comparison of group means difficult. We only included cases where the actual diagnosis of EIMs and IBD was clearly stated in patient history.

**Table 8 jcm-10-05984-t008:** The sequential order of extraintestinal symptoms in patients with more than one EIM.

Crohn’s Disease Patients with More Than One EIM—Order of Different EIMs				
Type of EIM	First EIM	Second EIM	Third EIM	Fourth EIM
Peripheral arthritis	19	18	1	0
Sacroiliitis	10	13	3	0
Spondylitis	2	6	2	0
Erythema nodosum	8	2	1	0
Pyoderma gangrenosum	2	2	1	0
Scleritis	0	1	0	0
Iridocyclitis	1	2	0	0
Ulcerative Colitis Patients with Multiple EIMs—Order of Various EIMs				
Type of EIM	First EIM	Second EIM	Third EIM	Fourth EIM
Peripheral arthritis	12	3	0	1
Sacroiliitis	2	11	1	0
Spondylitis	0	4	2	0
Erythema nodosum	2	0	1	0
Pyoderma gangrenosum	1	1	0	0
Scleritis	1	1	0	0
Iridocyclitis	2	1	0	0

Musculoskeletal findings were the most prevalent. Note that axial joint involvement usually developed later than peripheral arthritis. We only list the most common entities found in patients with multiple EIMs.

## Data Availability

The data that support the findings of this study are available from the corresponding author (I.F.) upon reasonable request.

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
