# Peer review of "Temporal Relationship of Extraintestinal Manifestations in Inflammatory Bowel Disease"

_jcm, 2021, doi:10.3390/jcm10245984_

Round 1
Reviewer 1 Report
The paper improved tremendously after revisions. It is in my opinion publishable.
Author Response
Response to Reviewer 1
We are sincerely grateful for your reviewing and commenting on our paper. We were delighted to hear, that you found, our work improved in comparison to earlier submitted version.
Sincerely yours
The authors

Reviewer 2 Report
The article already underwent various rounds of peer review.
I think that the article is already eligible to be published.
Author Response
Response to Reviewer 2
We would like to express our gratitude for your work on reviewing our manuscript. We were delighted to hear, that you found, our paper is eligible for publishing.
Sincerely yours
The authors
Reviewer 3 Report
In this study, Istvan Fedor, et al. characterized the temporal onset of extraintestinal manifestations (EIM) in inflammatory bowel diseases (IBD).
The results are well described. However, the study does not provide much novel information than the existing literature.
Authors may want to consider exploring temporal relationship between EIM and UC subgroups (ulcerative proctitis, left sided and pancolitis) and severity of IBD, and study risk factors for development of EIM.
I would suggest to rewrite the discussion part since majority of discussion is the repetition of the results. I would also recommend to discuss the limitations of the study in the later part of the discussion.
Author Response
Response to Reviewer 3:
First of all: We are grateful for your reviewing and commenting on our manuscript. We would also like to express our gratitude for the suggestions made. We attempted to address the raised points, and correct the manuscript accordingly.
We believe, the new version of our paper displays several novelties in comparison to the former version. From the previous version, we would like to highlight that EIM-s can occur anytime along the disease course, that Crohn’s disease patients are equally prone to developing EIM-s, regardless of the involved bowel segment, and also, that there is no observable difference in the prevalence of EIM-s in CD and UC.
In accordance with your suggestions, we evaluated the affected colon segments in UC (similar to keeping record of involved segments in CD). We also recorded the onset of EIM-s in different subgroups, as suggested. We included the new data and findings in the revised version
The authors admit, that, your raised concern about the discussion is valid and reasonable. Thereby, we aimed for a correction of the discussion section, in order to make the paper more concise and less redundant.
We again are sincerely thankful for your valuable comments and suggestions for improvement. We hope, the revised version of our manuscript got better in both quality and readability.
Sincerely yours
The Authors

Round 2
Reviewer 3 Report
Authors addressed all my concerns.
This manuscript is a resubmission of an earlier submission. The following is a list of the peer review reports and author responses from that submission.
Round 1
Reviewer 1 Report
This retrospective study includes 508 pat treated at one clinical center and aims to characterize the patient population based on data from “ medical documentation “ . No information is provided about the time frame.
General characteristics are recorded including age at diagnosis, location and behavior of disease. An interesting finding is the high prevalence of B3 in patients diagnosed at young age although the number of patients is very low, probably too low to permit a real conclusion. It remains unclear if behavior was determined at diagnosis or during disease progression.
Authors report in the introduction “ inconsistent data about the frequencies of EIM “although a lot of data are reported in the literature with a consensus meting at ECCO. In table 4 EIM are reported although a lot of pathologies can not be considered as EIM : cholelithiasis – conjunctivitis – pancreatitis ( only autoimmune pancreatitis is EIM) - primary biliary cholangitis . In contrast uveitis and episcleritis are not reported. ( cfr ECCO consensus ). Correlation and comparison with data from the literature are flawed. Authors don’t justify their interest in the temporal relationship between EIM and diagnosis of IBD neither their interest in the sequence of multiple EIM’s.
The authors claim “ We also conducted a non-comprehensive literature search in Pubmed, seeking previous results in the characteristics of extraintestinal manifestations of these diseases focusing on more recent literature. The comparison of our results with previous reports was also among our aims “ . The result is weak and poor.
In general the study is not innovative and the text is sloppy with sometimes a lot of irrelevant details ( for ex; use of determination of bile acids in serum to differentiate CD and UC; gender differences in several items.. ). Language needs extensive editing.
Author Response
Response to reviewer 1:
First, we would like to express our appreciation for reviewing our writing. Thank you for the valuable points and remarks, and also for your time. In the revised version of the manuscript, we attempted to resolve the raised issues accordingly.
This retrospective study includes 508 pat treated at one clinical center and aims to characterize the patient population based on data from “ medical documentation “ . No information is provided about the time frame.
In the revision, we included the time frame, in which we assessed health records of the patients (2008-2021).
General characteristics are recorded including age at diagnosis, location and behavior of disease. An interesting finding is the high prevalence of B3 in patients diagnosed at young age although the number of patients is very low, probably too low to permit a real conclusion. It remains unclear if behavior was determined at diagnosis or during disease progression.
When considering Montréal classification of patients, we determined the length of inflamed bowel and disease behavior during the disease course, not at initial presentation.
Authors report in the introduction “ inconsistent data about the frequencies of EIM “although a lot of data are reported in the literature with a consensus meting at ECCO. In table 4 EIM are reported although a lot of pathologies can not be considered as EIM : cholelithiasis – conjunctivitis – pancreatitis ( only autoimmune pancreatitis is EIM) - primary biliary cholangitis . In contrast uveitis and episcleritis are not reported. ( cfr ECCO consensus ). Correlation and comparison with data from the literature are flawed. Authors don’t justify their interest in the temporal relationship between EIM and diagnosis of IBD neither their interest in the sequence of multiple EIM’s.
Thank you for pointing that ECCO consensus guideline lists extraintestinal manifestations. Accordingly, we removed entities, that cannot be considered as EIM-s of inflammatory bowel diseases. We would also like to justify our interest in the temporal relationship between EIM-s and the diagnosis of IBD. Assuming the role of common pathomechanisms in IBD and EIM, their sequencing may also be significant. Perhaps it is worth paying attention to this in other specialities.
The authors claim “ We also conducted a non-comprehensive literature search in Pubmed, seeking previous results in the characteristics of extraintestinal manifestations of these diseases focusing on more recent literature. The comparison of our results with previous reports was also among our aims “ . The result is weak and poor.
In general the study is not innovative and the text is sloppy with sometimes a lot of irrelevant details ( for ex; use of determination of bile acids in serum to differentiate CD and UC; gender differences in several items.. ). Language needs extensive editing.
We have to agree with you, that our data hardly presents any novelties. As neither of the authors is a native speaker of English, we expected, that language fluency might not be flawless. Thereby, we opted for a professional language correction of our corrected manuscript, ordered through the website of the Journal of Clinical Medicine - MDPI.
We also aimed for reducing unnecessary information, thus we deleted sections on serum bile acid measurements and anti-saccharomyces cerevisiae antibodies. We also removed redundant parts about gender differences in the revised version.
We kindly appreciate your valid and reasonable remarks. We revised our writing accordingly, and hope, the quality of our paper improved. Thank you for your time and effort.
Reviewer 2 Report
An interesting original study reporting extraintestinal manifestations in patients affected by Inflammatory Bowel Diseases.
I have some questions:
Please add the exact version of Medcaalc software and the other software utilized, as well as the maker and location.
Please add what kind of t-test you performed (paired, unpaired?)
Page 3 line 91: "3.2. Crohn’s Disease Patients Presented Younger" please check English!!
line 208 you should add: "erythema nodosum is also more frequently associated with Crohn's Disease" and cite an article such as: doi: 10.1111/dth.12811.
Regarding the side effect of psoriasiform reactions (5 cases), you reported, how many of those patients were under biological drugs? Psoriasiform reactions (if psoriasis was not present before starting biological drugs) are very common in those particular subtypes of patients. you should add this information and a citation, such as: doi: 10.1080/03007995.2020.1786681.
In the limitations of the study, you should state that you did not consider the treatment of patients, as many side effects may be also associated with drug intake (see psoriasiform reaction).
Thank You
Thank You
Author Response
Response to Reviewer 2:
We are grateful for reviewing and commenting on our manuscript. We revised the writing in order to improve the readability and quality.
An interesting original study reporting extraintestinal manifestations in patients affected by Inflammatory Bowel Diseases.
I have some questions:
Please add the exact version of Medcaalc software and the other software utilized, as well as the maker and location.
Thank you for pointing this out. We added both the versions of Medcalc, and the edition of Microsoft Excel.
Please add what kind of t-test you performed (paired, unpaired?)
We added details on the t-test used. The t-tests were for independent samples, thereby, unpaired, and we used two-tailed t-test to determine whether the groups differ.
Page 3 line 91: "3.2. Crohn’s Disease Patients Presented Younger" please check English!!
We revised this heading, we hope it improved the text quality.
line 208 you should add: "erythema nodosum is also more frequently associated with Crohn's Disease" and cite an article such as: doi: 10.1111/dth.12811.
Indeed this was a missing information, thereby we included it in the revised manuscript. Also added citation [1].
Regarding the side effect of psoriasiform reactions (5 cases), you reported, how many of those patients were under biological drugs? Psoriasiform reactions (if psoriasis was not present before starting biological drugs) are very common in those particular subtypes of patients. you should add this information and a citation, such as: doi: 10.1080/03007995.2020.1786681.
Thank you for pointing out this issue. We revised the paper accordingly, mentioning the possible psoriasiform eruptions with biologics, particularly Adalimumab. Also added citation on a paper assessing the safety profile of different biological agents [1,2].
In the limitations of the study, you should state that you did not consider the treatment of patients, as many side effects may be also associated with drug intake (see psoriasiform reaction).
In the revised version, we added to the limitations, that GI specialists did not treat the extraintestinal findings of patients, as these were managed by other professionals.
We are again grateful for reviewing our manuscript. We would also like to express our gratitude for the suggestions made. We attempted to address the raised points, and correct the manuscript accordingly. We hope, the revised version of our manuscript had improved in quality.
Sincerely yours
- Spagnuolo R, Dastoli S, Silvestri M et al. Anti-interleukin 12/23 in the treatment of erythema nodosum and Crohn disease: A case report. Dermatol Ther 2019;32:2–3.
- Roberti R, Iannone LF, Palleria C et al. Safety profiles of biologic agents for inflammatory bowel diseases: a prospective pharmacovigilance study in Southern Italy. Curr Med Res Opin 2020;36:1457–63.
Round 2
Reviewer 2 Report
The paper showed an overall improvement, also considering english. It is in my opinion publishable.